# IS TWO ALWAYS BETTER THAN ONE? Customer Perception on the Merger of Startup Decacorn Companies

Ilham Fauzan Putra *, Nila Armelia Windasari, Gita Hindrawati and Prawira Fajarindra Belgiawan

School of Business and Management, Institut Teknologi Bandung, Bandung 40132, Indonesia;
nila.armelia@sbm-itb.ac.id (N.A.W.); gita_hindrawati@sbm-itb.ac.id (G.H.); fajar.belgiawan@sbm-itb.ac.id (P.F.B.)
* Correspondence: ilham_fauzan@sbm-itb.ac.id

**Abstract:** Recently, the two decacorn Startups in Indonesia decided to collaborate with a merger scheme and accumulate over 100 million monthly active users. The Merger triggers a new perception for customers and has an impact on the level of Brand Loyalty. However, no research examines how customer perception to mergers between the startup with decacorn status. Therefore, this study intends to fill this gap. Using the mix method approach, this study investigates how customer perception of startups' mergers decacorn status and examines its effect on customer behavior. The findings revealed that customer self-congruency successfully mediates the research gap between customer perceptions on merger scheme (sig. $0.000 < 0.05$, $\beta = 0.575$) and purchase intention (sig. $0.000 < 0.005$, $\beta = 0.390$) by combining impulse buying strategies (sig. $0.003 < 0.05$, $\beta = 0.329$). The paper contributes theoretically to the body of knowledge in the customer's perception of the merging company. The study also gives new insight that customers' perception of the Merger of two decacorn companies cannot guarantee that customers will be loyal to the company without connecting the customer self-congruency in each partner. It is recommended that the manager gain stimulus in building conformity of company image with customer perceptions that create self-congruency.

**Keywords:** brand loyalty; customer self-congruency; customer perception; impulse buying; merger and acquisition; purchase intention; startup

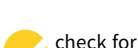



## 1. Introduction

The growth of startup companies in Indonesia continues to experience rapid progress. It is proven that Indonesia currently occupies the 5th position globally with 2.193 startup companies [1]. Currently, Indonesia has two startup companies with decacorn status. The companies are company X as mobile on-demand services, and payments platform contributes IDR 152 trillion (1%) to Indonesia's GDP in 2019. Furthermore, company Y is a technology company, especially in the e-marketplace, that contributes IDR 222 trillion (1.5%) to Indonesia's GDP in 2019. Recently, the two companies collaborated with a merger scheme and accumulated over 100 million monthly active users. Mergers and acquisitions, or M&A, are prominent strategic business options around the world [2]. Merger and acquisition (M&A) success is determined not only by the interests of both parties but also by relationships with other stakeholders, such as local government and public recognition, and financial institution support [3]. Companies tend to focus on financial performance after completing the M&A agreement, but they are less concerned about how customers perceive the M&A [4]. At the same time, the Merger will trigger a new perception for customers and impact the level of loyalty to the brand. Therefore, companies need to know from the start how customer perception to the Merger [2]. Pre-merger service perceptions are likely to influence service evaluations once a merger is announced [2]. Although mergers and acquisitions (M&As) are crucial strategic decisions that firms make to acquire both tangible and intangible advantages [5], it turns out that less than 50% of mergers are successful [4]. Throughout the merger and acquisition process, the worst strategy for brand

managers is to do nothing after the Merger and let the brands continue their separate ways as before [6]. Therefore, managers must know how the differences between the two brand representations can affect acquirer and acquired brands [4] because a merger frequently impacts both brands' image [5].

Several studies have researched how customers respond to company M&A, including the impact on service perceptions and brand loyalty [2,5]. However, no research examines how customers' perception of mergers between startup companies with decacorn status. When the company's size gets bigger, it could change its brand landscape and its positioning in the market [7]. On the other hand, recently, there has been a significantly increasing number of startups and innovations across the globe. This study could potentially shed light on how startups, with their dynamic and aggressive market orientation, do corporate actions and affect how their customer perceives them.

This study set is in Indonesia, whereas as many as 88.1% of internet users in Indonesia make purchases online. Thus, this research is essential for the global market since Indonesia is the country with the fifth most significant number of startups globally and the first largest online consumer in the world [8]. Besides, there is a possibility that the results of previous studies cannot be applied in Indonesia due to cultural differences or the theory application gap [9]. Besides, there is no consistent result between the findings regarding the effect of customer perception of M&A on customer purchase intention [5,10,11].

This study investigates customer perception of the Merger of two startup companies with decacorn status and examines its effect on customer behavior. Also, this study intends to examine the possible mediating role in the model to fill the gap. This study provides a theoretical contribution about the impact of customer perceptions of the Merger conducted by decacorn startup on loyalty and purchase intention. This study succeeded in closing the contradictory gap between customer perceptions of the merger scheme on purchase intention through customer self-congruency and impulse buying. Furthermore, this study also provides practical contributions, which the results reveal that the user interface is crucial. The user interface is one of the company's image representations, which has proven to play an essential role in building customer self-congruence. Therefore, companies must create an attractive and user-friendly user interface.

This study is organized into six sections. Section 1 is an introduction to the Merger and acquisition of two decacorn companies, and other relevant aspects of the research topic are included. Section 2 explains the theoretical basis used, the conceptual model, and the hypotheses of this study. Section 3 describes the data collection and methodology used in this study. The research results will be displayed in Section 4. Furthermore, in Section 5, the theoretical contributions and implications for practice will also be discussed. Finally, the conclusions and limitations of the research will be discussed in the last section.

## 2. Literature Review

The current COVID-19 pandemic has harmed various sectors of the economy [12]. Companies must adapt to existing changes to survive and face challenges [13]. Open innovation has an essential contribution to business recovery during this pandemic [14]. Regardless of their size or influential industry, open innovation is increasingly a key predictor of corporate success, effective performance, and organization survival [15]. Open innovation in firms generally pursues three primary goals [16]. First is speed internal innovation processes, second is expanding market opportunities, and the last is helping companies generate profit [16]. Startups concentrated in the modern technology sector, such as IT, fintech, and telecommunications, are also affected by the COVID-19 pandemic situation [17]. To develop and refine the business model of a technology-based company, they adopt open innovation practices [18]. Technology, ideas, and knowledge that freely cross inside and outside can be captured by companies if they apply open innovation to their business [19]. Open innovation helps startups to be able to match the needs and desires of consumers [20]. In this sense, the customer becomes a source of open innovation since, through collaboration, they can communicate their needs to the business

network [21]. Startups can succeed in implementing open innovation through their web-
sites and technology platforms' security, design, and ease of use [22,23]. If the needs of
consumers can be met, this will impact satisfaction and loyalty [20]. Open innovation is
attached to mergers and acquisitions because this concept is related to utilizing external
knowledge for the company's internal use; thus, one of the open innovation concepts is
Merger and acquisition [24]. Mergers and acquisitions (M&A) are a prominent company's
strategic alternative globally [2]. Many companies overly focused on elements such as
legal and regulatory, financial, and lack of concern for consumer perceptions even though
consumer perceptions of mergers can influence their behavior [2]. The study relies on
the effect of customer perception on merger schemes between two decacorn startups in
Indonesia, X and Y company. In this section, the theoretical basis used in this research
is explained. Furthermore, this section also explains how hypotheses are developed. In
addition, the conceptual framework is presented in this section.

### 2.1. Customer Perception on Merger Scheme

Customer perception of a brand is a significant aspect in determining customer loyalty
and, as a result, sales [25]. Customer perceptions of mergers are important to study
because this Merger involves two or more companies to change customer perceptions of
the company's image. Initial customer perceptions regarding a merged brand are likely to
shift due to the merger announcement and the pre-merger brands' brand image valence
(positive or negative) [2]. On the other hand, customers may assess fit in horizontal mergers
based on the valence of the brands engaged in the Merger (positive or negative) [2]. In
other words, acquiring a brand with a superior image could increase loyalty for the new
brand rather than acquiring one with an average image [4]. In addition, the difference
between the variance of two brand images was the greatest for brand loyalty [4]. Pre-
merger service perceptions are likely to influence service evaluations once a merger is
announced. Consequently, the Merger will affect customer expectations of the merged
firm's service (service quality) [2].

### 2.2. Brand Loyalty

Brand loyalty measures a customer's attachment to a brand, and it motivates cus-
tomers to purchase a preferred brand consistently [26]. Greater brand loyalty leads to
better customer satisfaction, excellent brand choices, and expenditure [27]. Customers that
are emotionally loyal like and express satisfaction with the product, brand, or service [28].
Loyal customers are critical to the success of any firm. When circumstances are harsh and
competition is fierce, their value skyrockets [29]. Customers like this do not just appear
out of nowhere. On the contrary, it usually takes a lot of time and effort on the company
first to entice them and then continue to supply their needs [29]. Customers are more
likely to develop loyalty when they positively think about the company, product, and
brand. Therefore, it is essential to investigate how mergers and acquisitions affect brand
loyalty [5].

### 2.3. Purchase Intention

The intention of a customer to buy the product or use the service is referred to as
purchase intention [30,31]. According to the literature, purchasing intention has various
meanings. To begin with, it relates to the likelihood that customers may be "willing to
consider purchasing." After that, it indicates what a person "wants" to purchase in the
future. Lastly, it displays a customer's decision to "purchase" a company's product "again."
According to [32], it is a trading activity following a customer's overall product appraisal.
It is a perceptual reaction to one's attitude toward a particular thing. That is, customers'
purchasing intentions are shaped by their product evaluations or attitudes toward a brand,
as well as external stimuli. The current pandemic has led to an increase in online shopping
and food delivery [33]. Of course, this will impact how consumers evaluate the brand
before they intend to buy the product they need. In online transactions, consumers tend

to evaluate products through reviews from other consumers [34]. Products that have good reviews tend to increase consumer purchase intentions [34]. In addition, previous research stated that the vital information for the customer in purchase decision-making is the price [19]. Besides price, discounts are also a consideration for consumers in buying products [35]. Hawaldar et al. [35] revealed that consumers believe the price is an indicator of quality.

## 2.4. Customer Self-Congruency

Customer self-congruency is derived from self-congruence theory, which was first introduced by [36]. He modified self-concept theory from Levy [37] and focused on self-image [37,38], who revealed that every product has an image that does not only stop at the appearance of the product in a physical way, but also other factors such as price, packaging, or marketing [39]. The study of [36,40] sees that self-image on the product has a relationship with the self-concept that exists in customers. For instance, a product with a luxury image will affect the customer's identity to become a high-status person. Self-congruence stands between the self-concept of the customer and the image of the product itself. Refs. [41,42] emphasized that the congruity of the product image with the self-concept is crucial since it is a factor considered by customers before deciding to make a purchase. Based on the description above, it can be reaffirmed that customer self-congruence exists when there is a linear or congruent relationship between the image that a product wants to display and the target customer who will use it. Customers tend to form connections with products through physical and psychological to determine the actual desire felt by customers [43]. In the context of this study, customer self-congruency reflects the alignment of the connection felt by customers to the merger scheme carried out by two companies that have become part of customers' daily lives. Merger activity indicates a change in business processes within the company, both major and minor. This change will affect the customer's perspective and change the product self-image of the customer's self-concept.

## 2.5. Impulse Buying

According to [44], Impulse buying is a concept in marketing that describes buying activities that customers do not plan. More than that, impulse buying can occur when customers get a stimulus when shopping, which encourages them to make purchases spontaneously [45]. The concept of impulse buying is introduced first by [46], emphasizing its crucial role in dealing with dynamic markets. This concept is one of the strategies in conducting customer engagement. However, as [47], where the concept of impulse buying is contrary to Maslow's theory of motivation and the theory of needs, showing that marketers can change the orientation of customers who are generally considered rational when making purchases to become irrational. Purchasing decisions with impulses from external and affecting the self-internal of customers will turn into motivations contrary to rational thinking [48]. Impulse buying in the context of corporate merger activities shows how far customers' level of rationality in purchasing decisions can change when a stimulus appears in the form of merger activities between companies.

## 2.6. Customer Perceptions on Merger Scheme and Customer Self-Congruency

The activity of merging two companies into one is a business strategy to maintain a position in market competition [49]. However, the merger scheme is not only a business strategy in administration but also in marketing [50,51]. Merger schemes can be a marketing strategy to attract customers' attention because of unusual movements in a well-known company. The emergence of customer interest in the merger scheme results from the stimulus given to increase market expansion. Based on Self-Congruency theory, the merger scheme can trigger customers and non-customers who have linearity in each self-concept. For non-customers, when they hear about a merger scheme in a company that has congruence with the self-concept, it will give them an impetus to look at the company. As for the customers themselves, the merger scheme can affect the perception of changes

in the company. Customer perceptions, both positive and negative, will affect customer self-congruency [5,52]. It is because customers will reconsider regarding the alignment of self-concept with the image of the company through the perceptions that arise [53,54]. The proposed hypothesis is as follows:

**Hypothesis 1 (H1):** *Customer Perceptions on Merger Scheme has a significant effect on Customer Self-Congruency.*

### 2.7. Customer Self-Congruency and Purchase Intention

Purchase intention is a critical aspect that is considered in various market research because it reflects the possibility of purchases that arise by customers [55], be a significant factor in measuring success in managing a brand [56], and become a consideration in assessing the performance of marketing strategy [10]. According to [42], purchase intention can be increased by creating customer self-congruency on the products offered or the company's image. Merger and Acquisition (M&A) carried out by the company will impact the image and brand identity [51]. If consumers feel that the image of the merged company does not match their perception, then they tend not to buy [51]. In line with [57] referring to the Self-congruity theory, the congruence of self-personality to the company's image significantly affects purchase intention. That is due to good information transfer of product knowledge to customers. A company image conveyed well to customers will stimulate the growth of the congruence process and impact the emergence of purchase intention [58]. The proposed hypothesis is as follows:

**Hypothesis 2 (H2):** *Customer Self-Congruency has a significant effect on Purchase Intention.*

### 2.8. Customer Self-Congruency and Brand Loyalty

It is unavoidable that the meaning of a brand will change due to significant changes at the corporate level, such as Mergers and acquisitions, which influence the existing relationship and connections people have with the brand. Due to M&A, a brand's identity may take on undesirable traits; for example, customers' reflections of their self- and ideal image through a brand may be distorted [59]. Self-congruence has been linked to the self and ideal image. Thus, it can be stated that self-congruence can be related to brand loyalty in the M&A scheme. In particular, brands have a "personality" that reflects the stereotypical picture of the brand's average user—the brand-user image [60]. Customers try to evaluate a brand by connecting the brand-user image (also known as "symbolic qualities") with their self-concept (actual self-ideal self and social self), according to [60]. According to self-congruence theory, when a customer's self-concept is consistent with a brand, the customer is more likely to create preferences, good feelings, and a sense of belonging to the brand [61]. The closer the brand-user picture resembles the customer's ideal self-image; the more likely customers would infer that using the brand will satisfy their self-esteem demands [60]. Several studies have found a direct correlation between self-congruity and loyalty [52]. According to the findings of [60], self-congruity predicted brand loyalty favorably and substantially. When a customer's self-image aligns well with the brand image, this is known as high self-congruity [62]. The customer will become loyal to the brand because of this. The solid conceptual support in related studies supports the expectation that when customers perceive a high level of alignment between themselves and the brand, they are more likely to develop a sense of belonging to and engagement with the brand. As a result, we argue that congruity catalyzes the customer's passion and immersion in the brand [61]. Therefore, the third hypothesis is formulated as follows:

**Hypothesis 3 (H3):** *Customer self-congruency has a significant effect on brand loyalty.*

### 2.9. Brand Loyalty and Purchase Intention

Purchase intention can be used to quantify the tendency of a customer to buy a product, the link between these two components is such that the higher the buying intention, the larger a customer's desire to buy a thing [63–65] defined brand loyalty as a desire to always repurchase or patronize a favorite product or service in the future, regardless of any marketing strategies or situational variables that can affect switching behavior. If consumers have a negative perception of a merger and acquisition (M&A) brand, then it can affect customer loyalty and have a greater intention to switch brands [66]. According to [67], regarding behavioral perspective, brand loyalty refers to the tendency of a buying unit, such as a household, to purchase the same brand in a product category over a specified length of time. Similarly, Ref. [65] defined brand loyalty as a customer's ability to remain loyal to a single brand. Despite competing businesses' marketing efforts, customers' intentions to purchase the brand as their first option indicate this behavior. According to [68], brand loyalty can affect customer decisions to buy the same product or brand and avoid switching to other brands. Furthermore, brand-loyal customers will buy the brand without hesitation based on previous experiences, even if they have not researched [69]. Thus, it is further hypothesis that:

**Hypothesis 4 (H4):** *Brand loyalty has a significant effect on purchase intention.*

### 2.10. Customer Self-Congruency and Impulse Buying

Companies frequently employ mergers and acquisitions to improve their performance and attract customers [70]. Customers tend to consider products from the physical and psychological sides in the purchase decision-making process to find congruence with themselves [40]. Likewise, companies that produce goods/services become a crucial aspect that affects product sales. So, self-congruency between customers and companies must be built to form engagement that impacts loyalty to the company [71]. However, for companies that want to increase sales significantly within a certain period, a particular strategy is needed to attract customers' attention, one of which is impulse buying [72]. Impulse buying encourages customers to make purchases without planning, with the stimulus given to customer preferences for a company. However, companies cannot quickly build customer perceptions to create results, such as purchase intention [5]. Therefore, it is necessary to have additional concepts that encourage customer perceptions, and psychological aspects of customer behavior are the key to creating perceptions through the process of engagement with the self-concept of the customer [48]. In this context, customer self-congruency is the right strategy to increase impulse buying [73], where the resulting congruence will make it easier for customers to be given a stimulus because of the element of trust in the company so that that impulse buying will be created. The proposed hypothesis is as follows:

**Hypothesis 5 (H5):** *Customer Self-Congruency has a significant effect on Impulse Buying.*

### 2.11. Customer Perception and Brand Loyalty

According to [74], an essential brand association is the overall brand attitude, which the customer's perception can conceive of the brand's product quality. According to [75], product quality perception is a broad assessment of the customer's perception of the product's superiority. It is vital to explore how customers deal with M&A identity dissonance induced by changes in brand meanings to produce more successful M&A results and better secure existing customers [76]. According to previous research, improved customer good perceptions lead to enhanced attitudinal loyalty. According to [77], increased satisfaction has a considerable impact on attitudinal and behavioral loyalty, i.e., a satisfied client would make repeat purchases and recommend the brand to friends. Ref. [78] also found that satisfaction and loyalty had a positive association. The study also revealed that brand loyalty grows when a customer has a good perception of and appreciates a particular brand [79]. Hence, the authors suggest the following hypothesized relationship:

**Hypothesis 6 (H6):** *Customer perception has a significant effect on brand loyalty.*

### 2.12. Brand Loyalty and Impulse Buying

Due to their variety-seeking tendency, compulsive buyers have a lower level of brand loyalty, whereas non-compulsive buyers trust their favorite brand and consider buying other brands to be riskier [80]. According to [81], compulsive purchasers prefer well-known and higher-priced brands because they give them recognition, make them feel better, and increase their self-esteem. Customers feel excitement and novelty in their shopping experience with variety seeking and impulse buying, according to [82], and it provides a break from boredom. Brand loyalty, on the other hand, refers to a customer's level of commitment to a particular brand [83,84]. According to [85], brand loyalty refers to the consistent purchase of a single product toward which the buyer has a favorable opinion. In a similar vein, Ref. [28] defined brand loyalty as the purchase of a single brand regularly. However, Ref. [86] discovered that brand loyalty has a positive impact on impulsive buying. Based on the above discussion, the following hypothesis is proposed:

**Hypothesis 7 (H7):** *Brand loyalty has a significant effect on impulse buying.*

### 2.13. Impulse Buying on Purchase Intention

The concept of impulse buying has attracted the attention of many researchers in market research because of its vital role in influencing the level of purchase intention [87,88]. During the post-merger acquisition integration stage, customers' purchasing intentions were directly influenced by perceived changes in service quality and product variety [89]. Customers tend to have purchase intentions when they feel that the realization process is easy, safe, and can be done immediately [87]. This implementation process can be done immediately through impulse buying activities because of the spontaneity aspect given to customers. Impulsive buying behavior also often leads to the emergence of psychological effects such as feeling happy when shopping for the desired product immediately [90]. Thus, it can be concluded that the relationship between impulse buying done spontaneously is in line with the purchase intention that is present because of the fast and safe process. The proposed hypothesis is as follows:

**Hypothesis 8 (H8):** *Impulse Buying has a significant effect on Purchase Intention.*

Departing from the explanation above by drawing a relationship line based on the previous literature, conceptual models and hypotheses built in this study can be seen in Figure 1.

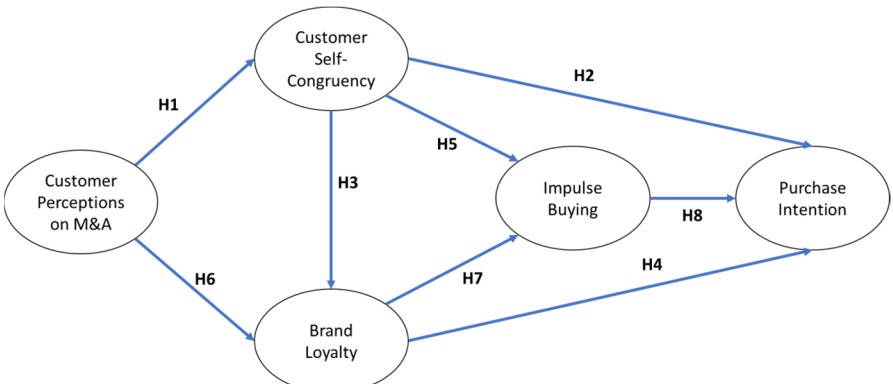

**Figure 1.** Conceptual Model. Source: Own Study.

## 3. Materials and Methods

The approach used in this paper is a mixed-method through qualitative (study 1) and quantitative methods (study 2).

### 3.1. Study 1

The first study used the qualitative method. The purpose of study 1 is to determine the customer perceptions of the merger scheme carried out by two decacorn companies and look at possible aspects that were influenced by customer perceptions. Based on the literature study, it is found that two variables are closely related to customer perceptions, as follows: brand loyalty and purchase intention, but it is possible to obtain new variables that support the study.

The study employed direct qualitative research conducting a focus group discussion (FGD). FGD informants have used X and Y platforms to get more in-depth preferences regarding developments and changes that appear on both platforms. The list of questions that form the basis for conducting FGDs to obtain information can be seen in Table 1.

**Table 1.** List of Questions.

| No | List of Questions |
|----|-------------------|
| 1 | What is your favorite e-commerce? Why? |
| 2 | What is your favorite online transportation service? Why? |
| 3 | Are you familiar with the Y platform? Do you use Y platform's services? How often? What is your reason for using Y platform? What products do you buy the most at Y platform? |
| 4 | Are you familiar with X platform? Do you use Y platform's services? How often? What is your reason for using X platform? What services do you use most often? |
| 5 | What do you think about Y platform? |
| 6 | What do you think about X platform? |
| 7 | In your opinion, are these two platforms easy to use? |
| 8 | What was your opinion when X and Y platform decided to merge and become one under a company? If it does not match, what are the reasons? How it should be? Are there other brands that should collaborate? |
| 9 | What kind of service do you expect from this Merger? |
| 10 | After this Merger, has your perception changed towards these two brands? What is the reason? |
| 11 | For infrequent users: after knowing the information about this Merger, are you interested in using X and Y platforms? |
| 12 | What e-wallet do you usually use? |
| 13 | Y platform plans to include an e-wallet of the partner on its platform; what do you think? |
| 14 | If you are a user of an e-wallet from Y platform, what do you think? Will you choose to keep using it or change to the X platform's e-wallet? |

Source: Own Study.

The FGD was conducted online via a video teleconference held on the Google Meet application on Sunday, 13 June 2021. The FGD consisted of two moderators and five participants. It took approximately one and half hours. There are five informants, of which two (40%) are postgraduate students, and three (60%) are already working. They are chosen as the informant following a random sampling method. The total of the informant has met the minimum requirement based on the study of [91]. The detailed description of each informant can be seen in Table 2.

**Table 2.** Description of Informants.

| Informant Code | Gender | Age (Years Old) | Occupation |
|----------------|--------|-----------------|------------|
| IC1 | Male | 26 | Entrepreneur |
| IC2 | Male | 26 | Private employee |
| IC3 | Female | 26 | Master Student |
| IC4 | Female | 24 | Master Student |
| IC5 | Female | 24 | Entrepreneur |

Source: Own Study.

At the beginning of FGD, the moderator thanked participants for their willingness to participate in the FGD. After that, the moderator explained the purpose and objectives of the FGD. Further, ask participants for permission to record this FGD. Then, the moderator asks the respondent questions; then, the respondent is asked to answer. Participants will first be appointed to answer questions, but there are also sessions where participants should raise their hands to answer. At the end of the session, the moderator gave a closing statement about the discussion and thanked the participants for being willing to participate in the FGD. It ended when the answers were already saturated with the throughout-the-topic discussion.

### 3.2. Study 2

The second study aims to confirm the model built in study 1 and answer study questions related to the relationship between Customer Perception of Merger Scheme and Purchase Intention. In addition, this study aims to examine the role of Customer Self-Congruency as a mediating variable in bridging the relationship between Customer Perception of Merger Scheme and Purchase Intention.

The second research method uses a quantitative approach by distributing questionnaires using a random sampling method from June–July 2021. Every construct is represented by at least three indicators measured by a 1–7 Likert scale. The indicators that are built based on previous studies on each variable can be seen in Table 3.

**Table 3.** Measurement of Variable.

| Variable | Measurement Items |
|---|---|
| **Customer Perception on M&A [5]** | |
| CP1 | This Merger is good. |
| CP2 | I am glad to hear about this Merger. |
| CP3 | This Merger will be helpful. |
| **Customer Self-Congruency [61,73]** | |
| CSC1 | The user interface of X platform has an image similar with how I like to see myself. |
| CSC2 | The user interface of Y platform has an image similar with how I like to see myself. |
| CSC3 | The user interface of X platform is consistent to how I like to see myself. |
| CSC4 | The user interface of Y platform is consistent to how I like to see myself. |
| CSC5 | I feel like I am part of X platform. |
| CSC6 | I feel like I am part of Y platform. |
| CSC7 | The user interface of X platform is consistent to how I would like other to scheme. |
| CSC8 | The user interface of Y platform is consistent to how I would like other to scheme. |
| CSC9 | People who use the X platform are very similar to me. |
| CSC10 | People who use the Y platform are very similar to me. |
| **Impulse Buying [72]** | |
| IB1 | When I know the merger scheme, I buy the things I had not intended to shop in X platform. |
| IB2 | When I know the merger scheme, I buy the things I had not intended to shop in Y platform. |
| IB3 | When I know the Merger of Go To, I makes unplanned purchase in X platform. |
| IB4 | When I know the Merger of Go To, I makes unplanned purchase in Y platform. |
| **Brand Loyalty [5]** | |
| BL1 | I will be committed to X platform. |
| BL2 | I will be committed to Y platform. |
| BL3 | I will continue to use X platform. |
| BL4 | I will continue to use Y platform. |
| BL5 | I will remain loyal to X platform. |
| BL6 | I will remain loyal to Y platform. |
| **Purchase Intention [92]** | |
| PI | When I know the merger scheme, I consider buying products in X platform. |
| PI | When I know the merger scheme, I consider buying product in Y platform. |
| PI | When I know the merger scheme, I intend to buy product in X platform. |
| PI | When I know the merger scheme, I intend to buy product in Y platform. |
| PI | When I know the merger scheme, I probably will buy product in X platform. |

Source: Own Study.

### 3.3. Pilot Study

The pilot test was conducted to test the validity and reliability of each variable. That is, at the beginning of the study as a preliminary study. The pilot study is to ensure that the selected indicators can appropriately represent the variables. This process is conducted to get the first alert if samples need to be revised before distributing to the large sample. It can reduce extra time and cost because of the error possibility. The pilot test can be carried out with a minimum sample of 30, and this study has met the minimum requirement of 38 samples [91].

On the pilot result, one of the Impulse Buying indicators (IB5) is deleted since the significance value (0.170) > alpha (0.05). Other variables suggested to be deleted, such as CSC10, BL6, and PI2, are still considered indicators. This is because the indicators are significant to measure the variable. Besides, all the suggested deleted indicators are valid since significance value < 0.05 and reliable with Cronbach Alpha's value > 0.7 [93]. After deleting indicators on IB5, all indicators are valid and reliable, as shown in Table 4.

**Table 4.** Second Validity and Reliability Test.

| Variable | Code | Validity | | Reliability | |
|---|---|---|---|---|---|
| | | Pearson Correlation | Significance | Reliability If Item Deleted | Cronbach Alpha |
| Customer Perceptions | CP1 | 0.811 | 0.000 | 0.768 | |
| | CP2 | 0.865 | 0.000 | 0.723 | 0.805 |
| | CP3 | 0.873 | 0.000 | 0.702 | |
| Customer Self-Congruency | CSC1 | 0.513 | 0.001 | 0.834 | |
| | CSC2 | 0.598 | 0.000 | 0.824 | |
| | CSC3 | 0.634 | 0.000 | 0.820 | |
| | CSC4 | 0.608 | 0.000 | 0.823 | |
| | CSC5 | 0.840 | 0.000 | 0.794 | |
| | CSC6 | 0.763 | 0.000 | 0.806 | 0.836 |
| | CSC7 | 0.720 | 0.000 | 0.811 | |
| | CSC8 | 0.631 | 0.000 | 0.821 | |
| | CSC9 | 0.605 | 0.000 | 0.828 | |
| | CSC10 | 0.511 | 0.001 | 0.843 | |
| Impulse Buying | IB1 | 0.907 | 0.000 | 0.829 | |
| | IB2 | 0.863 | 0.000 | 0.858 | 0.888 |
| | IB3 | 0.837 | 0.000 | 0.872 | |
| | IB4 | 0.854 | 0.000 | 0.865 | |
| Brand Loyalty | BL1 | 0.771 | 0.000 | 0.806 | |
| | BL2 | 0.793 | 0.000 | 0.801 | |
| | BL3 | 0.801 | 0.000 | 0.802 | 0.840 |
| | BL4 | 0.828 | 0.000 | 0.792 | |
| | BL5 | 0.707 | 0.000 | 0.838 | |
| | BL6 | 0.698 | 0.000 | 0.850 | |
| Purchase Intention | PI1 | 0.930 | 0.000 | 0.970 | |
| | PI2 | 0.881 | 0.000 | 0.976 | |
| | PI3 | 0.944 | 0.000 | 0.968 | 0.973 |
| | PI4 | 0.973 | 0.000 | 0.963 | |
| | PI5 | 0.961 | 0.000 | 0.965 | |
| | PI6 | 0.957 | 0.000 | 0.966 | |

Source: Own Study.

Considering that all indicators have been declared valid and reliable, the distribution of the questionnaire was continued and collected amount to 138 samples. However, eight samples were eliminated because they did not pass the screening question.

Table 5 displays that most respondents are female (60%), mainly in the age range between 19 and 25 years old (76.92%). It represents that the X and Y Platform is more attention from the female customer than males. Regarding the frequency of use, X Platform is eminent, with 40% of respondents using the platform more than four times. However,

most of the respondents (68.46%) in Y Platform only using the platform amount to 1–2 times per month. It can show the popularity of the platform within respondents that X company is ranked first.

**Table 5.** Characteristic of Respondent.

| | Total | Percent |
|---|---|---|
| Gender | | |
| Male | 52 | 40% |
| Female | 78 | 60% |
| Age | | |
| 19–25 years old | 100 | 76.92% |
| 26–32 years old | 25 | 19.23% |
| 33–38 years old | 2 | 1.54% |
| >39 years old | 3 | 2.31% |
| Frequency of Use of X Platform | | |
| 1–2 times | 54 | 41.54% |
| 3–4 times | 24 | 18.46% |
| More than four times | 52 | 40% |
| Frequency of Use of Y Platform | | |
| 1–2 times | 89 | 68.46% |
| 3–4 times | 15 | 11.54% |
| More than four times | 26 | 20% |

Source: Own Study.

## 4. Results

### 4.1. Result of Study 1

The data from the FGD is then analyzed using open coding, axial coding, and selective coding. Based on the results of observations in FGDs, the factors influenced by customer perceptions can be seen in Table 6.

Table 6 shows that the results of the FGD narrowed down to five main themes, namely Customer Perception of Merger Scheme, Brand Loyalty, Customer Self-Congruency, Impulse Buying, and Purchase Intention. The three variables are in line with the theory based on the literature study, namely Customer Perception of Merger Scheme [5], Brand Loyalty [94], and Purchase Intention [90]. There are two added variables as findings from FGD that is Customer Self-Congruency [40] and Impulse Buying [46].

### 4.2. Result of Study 2

This study uses PLS-SEM to analyze the data. PLS-SEM is a multivariate technique that can be used with small samples. The data collected from this questionnaire were 138 respondents. However, eight respondents were not included in the analysis stage because they did not meet the requirements of this study. Therefore, the sample of this study was 130. Outer loadings, reliability, convergent validity, and discriminant validity should be considered when analyzing the measurement model.

The first step in the analysis using PLS-SEM is to ensure the value of outer loadings. It is to confirm that all of the study model's elements contributed significantly to each construct assigned. If the value is close to or greater than 0.7, the item is retained [95]. If the value of outer loadings is between 0.4 and 0.6, considered for removal from the scale if the indicator increases the composite reliability (or the Average Variance Extracted) above the proposed threshold value. Values of outer loadings below 0.4 (very low) were eliminated from the scale. Based on the result analysis, the value of the outer loading of each item is close to or above 0.7.; it can be seen from Table 7.

**Table 6.** Content Analysis.

| Themes | Findings | |
| --- | --- | --- |
| | **Y Company** | **X Company** |
| **Main Themes** | | |
| Customer Perception of Merger Scheme | The informants already know information related to the Merger between X and Y company, but the new mechanism is still a question for them. They think that the Merger means that there is a possibility that a platform will emerge for them but there is no statement about this case. One of the informants also said that if the two big platforms in Indonesia were to merge, it would gather superpower in the market. Some of the participants also asked why X company did not cooperate with the competitor, the e-marketplace with the largest market share in Indonesia but instead chose Y company. The existence of e-wallet in Y, company provides customers with additional options for payment methods without leaving e-wallet in X company | |
| Brand Loyalty | It can be said that the platform is used frequently, but every time they make a purchase decision, almost all the informants compare the prices and discounts offered. | One of the informants is a loyal user, but after the platform update, the user switched to another platform. Besides, two of them still using the platform but rarely. An informant is using the platform periodically to get raw material in the subscription store. An informant uses almost all the features frequently. |
| Customer Self-Congruency | X company, as a young entrepreneur in Indonesia builds it, has its value for informants. However, the use of features on X company, which is limited to ride transportation/food delivery, makes its position on par with competitors. | It has a more complicated interface compared to other platforms. Partner shops and products displayed are still limited. Besides, the prices and discounts offered are considered unable to beat its competitors. However, one of the informants thought that the user interface was friendly and had exclusive features apart from shopping. |
| Impulse Buying–Purchase Intention | Before deciding to stop at the platform, the informants want to see the direction of this Merger and what benefits can be obtained. | |
| **Support Themes** | | |
| Brand Familiarity | All informants are familiar with Y Company | All informants are familiar with X. |
| Position brand compared to competitors | There are disparate answers. One of the informants chose X company as a favorite platform but primarily only for package delivery. Two of them said that, evenly, another platform outside the survey better than X company regarding the price—food delivery or ride transportation. However, most of them decide to use the platform based on price, distance, and service availability. | Company Y is a favorite platform of two informants and the only used platform for an informant. Three of them chose another platform outside the survey as their favorite platform |
| Service Preference | Food delivery, ride transportation, and package delivery. However, most of the informants always compared X company with the competitor based on the price to decide the service they want to get. | Lifestyle, shopping for raw materials, office equipment, gold investment, and mutual funds. |
| Value-added Expectation | <ul><li>Low—Competitive Price</li><li>More discounts, promotions, and cashback</li><li>Focus on the instant delivery</li><li>Easy and Friendly user interface</li><li>Cooperate with more partners, so more product choices to compare.</li><li>In environmental concern, it is hoped that there will be a feature to determine the wasted carbon emission compared to the distance of the package delivery.</li></ul> | |

Source: Own Study.

**Table 7.** Results of The Measurement Model Analysis.

| Construct | Outer Loadings | Cronbach Alpha's | Composite Reliability | Average Variance Extracted (AVE) |
|---|---|---|---|---|
| Customer Perception on M&A | | 0.851 | 0.910 | 0.771 |
| CP1 | 0.877 | | | |
| CP2 | 0.898 | | | |
| CP3 | 0.858 | | | |
| Customer Self-Congruency | | 0.894 | 0.919 | 0.517 |
| CSC1 | 0.638 | | | |
| CSC2 | 0.624 | | | |
| CSC3 | 0.756 | | | |
| CSC4 | 0.766 | | | |
| CSC5 | 0.837 | | | |
| CSC6 | 0.773 | | | |
| CSC7 | 0.779 | | | |
| CSC8 | 0.677 | | | |
| CSC9 | 0.693 | | | |
| CSC10 | 0.608 | | | |
| Brand Loyalty | | 0.867 | 0.901 | 0.603 |
| BL1 | 0.809 | | | |
| BL2 | 0.844 | | | |
| BL3 | 0.769 | | | |
| BL4 | 0.809 | | | |
| BL5 | 0.699 | | | |
| BL6 | 0.720 | | | |
| Impulse Buying | | 0.902 | 0.932 | 0.774 |
| IB1 | 0.897 | | | |
| IB2 | 0.804 | | | |
| IB3 | 0.906 | | | |
| IB4 | 0.903 | | | |
| Purchase Intention | | 0.947 | 0.958 | 0.791 |
| PI1 | 0.901 | | | |
| PI2 | 0.808 | | | |
| PI3 | 0.906 | | | |
| PI4 | 0.906 | | | |
| PI5 | 0.906 | | | |
| PI6 | 0.906 | | | |

Source: Own Study.

Furthermore, the Cronbach alpha and composite reliability values were used to measure internal consistency reliability. It can be seen in Table 7 that the value of the Cronbach alpha of each variable exceeds the threshold of 0.6 [95]. In addition, the composite reliability value of each variable also exceeds the threshold value of 0.7. Thus, it can be concluded that the model in this study is reliable.

In addition, convergent and discriminant validity values can be seen to measure the validity of the measurement in this study. Convergent validity can be seen from the Average Variance Extracted (AVE) value. The AVE rating criteria is more than 0.5 [95]. When predicting the dependent variable, discriminant validity is another validity measurement that evaluates how different a latent variable is from other latent variables [95]. Examining the correlation matrix among constructs was one prominent strategy for assessing the current study's discriminant validity. Ref. [77] stated that the AVE of any latent construct should be greater than the construct's highest squared correlation with any other latent construct. The square roots of all constructions' AVE values are calculated, and correlations

between constructs are analyzed. It can be seen in Table 8 that all constructs in this study reached this criterion.

**Table 8.** Discriminant Validity.

|  | Brand Loyalty | Customer Perception on M&A | Customer Self-Congruency | Impulse Buying | Purchase Intention |
|---|---|---|---|---|---|
| Brand Loyalty | 0.777 |  |  |  |  |
| Customer Perception on M&A | 0.471 | 0.878 |  |  |  |
| Customer Self-Congruency | 0.647 | 0.575 | 0.719 |  |  |
| Impulse Buying | 0.473 | 0.307 | 0.497 | 0.880 |  |
| Purchase Intention | 0.577 | 0.447 | 0.543 | 0.607 | 0.890 |

Source: Own Study.

Moreover, the coefficient of determination ($R^2$) is used to measure the combined effects of the exogenous latent factors on the endogenous latent variable. The higher the $R^2$ value, can boost the likelihood of correct predictions. An $R^2$ of 0.02 to 0.12 is regarded as weak, whereas one of 0.25 or more is substantial [95]. It can be seen in Table 9 that all endogenous variables have values above 0.25, which means brand loyalty, customer self-congruency, impulse buying, and purchase intention are substantial.

**Table 9.** Coefficient of determination ($R^2$) values.

|  | R Square | R Square Adjusted |
|---|---|---|
| Brand Loyalty | 0.433 | 0.424 |
| Customer Self-Congruency | 0.331 | 0.326 |
| Impulse Buying | 0.287 | 0.276 |
| Purchase Intention | 0.491 | 0.479 |

Source: Own Study.

Besides, the significance of the path coefficients was evaluated using the bootstrapping technique using 5000 sub-samples. Furthermore, the results of hypotheses testing can be seen in Table 10.

**Table 10.** Hypotheses Testing.

|  | Hypotheses | Original Sample/β | *p*-Value | Decision |
|---|---|---|---|---|
| H1 | Customer Perception on M&A → Customer Self-Congruency | 0.575 | 0.000 | Accepted |
| H2 | Customer Perception on M&A → Purchase Intention | 0.163 | 0.164 | Rejected |
| H3 | Customer Self-Congruency M&A → Brand Loyalty | 0.562 | 0.000 | Accepted |
| H4 | Brand Loyalty à Purchase Intention | 0.287 | 0.020 | Accepted |
| H5 | Customer Self-Congruency M&A → Impulse Buying | 0.329 | 0.003 | Accepted |
| H6 | Customer Perception on M&A → Brand Loyalty | 0.148 | 0.104 | Rejected |
| H7 | Brand Loyalty → Impulse Buying | 0.261 | 0.026 | Accepted |
| H8 | Impulse Buying → Purchase Intention | 0.390 | 0.000 | Accepted |

Source: Own Study.

The estimated model based on the analysis is displayed below:

Table 10 and Figure 2 reveal that six hypotheses are accepted, and two hypotheses are rejected.

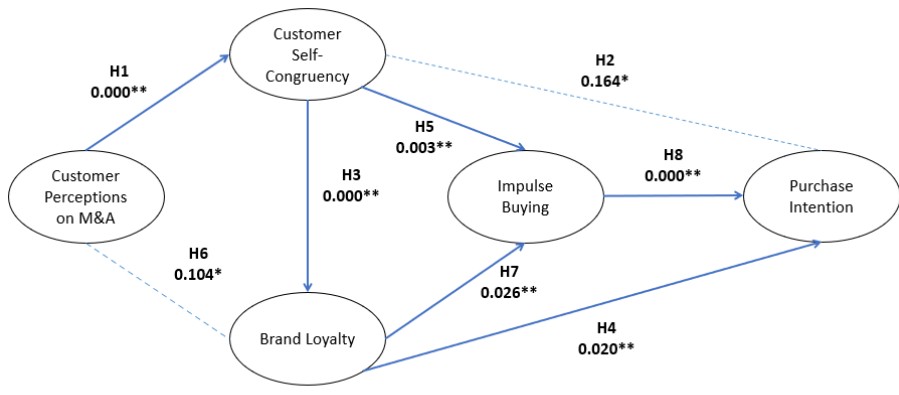

**Note:** **significant<0.05, *non-significant>0.05

**Figure 2.** Results of Hypotheses Testing. Source: Own Study.

## 5. Discussion

Based on the results of qualitative analysis, consumers have expectations of the mergers carried out by these startups companies. With this Merger, they hope to provide low-competitive prices, more discounts, promotions, and cashback. This result is in line with research from [19] which revealed that price and discount are consumer considerations in buying a product. Furthermore, consumers also expect an easy and friendly user interface. This expectation is one of the factors that can make the implementation of open innovation successful in startups [22,23]. In addition, consumers also hope that companies can collaborate with more partners to be compared when choosing products. The company can realize this expectation by conducting open innovation [21].

Furthermore, the result of the quantitative study shows that **H1 is accepted** (sig. 0.000 < 0.05, $\beta = 0.575$), which means that customer perception on merger scheme has a significant effect on customer self-congruency in line with [5,52,53]. It means that the customer's perception of the merger scheme is directly proportional to its self-concept. When X and Y companies merge, customers will reconsider the congruence between their identity and product image. Changes resulting from the merger process tend to change customer perceptions of the brand itself and shift the alignment of customer identity compared to corporate identity. Besides, regarding the acceptance of H1, customer perception affects the self-congruency, but there is also linearity in the image that he wants to show himself to others, resulting in firm engagement with high trust in the brand.

**H2 is rejected** (sig. 0.164 > 0.05, $\beta = 0.163$), meaning that customer self-congruency has no significant effect on purchase intention. This study contrasts with the findings from [96]; however, similar with a study from [57,58], where if product knowledge is not conveyed correctly to customers, then customers will have difficulty in capturing and interpreting the image raised by the company. That impacts the doubts that arise in the purchase decision-making process, thereby reducing the level of purchase intention. Thus, the presence of self-concept and product image congruence on the merger scheme is not enough to attract customers' attention and encourage their purchasing decisions. This relationship requires another concept that encourages customer perceptions to form self-identity and leads to purchase intention, such as impulse buying, which is supported by the acceptance of H5.

**H3 is accepted** (sig. 0.000 < 0.05, $\beta = 0.562$), meaning that customer self-congruency significantly affects brand loyalty. In line with a study from [97] that the high emotional attachment between consumers and a brand that can express their identity, brand loyalty is closely related to self-congruity. An inherent relationship between self-congruity and brand loyalty would further enhance the offerings' perspectives on products or services and help organizations retain the majority of their active consumers [98]. Moreover, the study results from [99] show that consumers who perceive the brand and their personality to be compatible would have stronger feelings towards their brand loyalty, then self-congruity

is the primary determinant of brand loyalty. The results of this quantitative analysis also confirm the findings we got when conducting FGDs where the identity suitability between customers and company mergers and the suitability of the service provider's user interface to the customer's wishes will create customer loyalty to the brand.

**H4 is accepted** (sig. 0.020 < 0.05, β = 0.287), meaning brand loyalty significantly affects purchase intention. This finding aligns with a study from [100], which stated a statistically significant effect on the relationship between brand loyalty and purchase intention. [101] show that the best indicator of purchase intention is brand loyalty. This study's results follow the statement of [102], which states that brand loyalty is an essential factor when customers intend to buy a product. Even though the brand has merged with other brands, loyal customers tend to have purchase intentions.

**H5 is accepted** (sig. 0.003 < 0.05, β = 0.329) and stated that customer self-congruency has a significant effect on impulse buying in line with [48,73]. It found that the provision of stimulus to customers must accompany that customer self-congruency to create impulse buying to give the impression that future purchases can be made quickly and safely. The similarity of characters shown by the X and Y platform through the user interface can trigger customers to make spontaneous purchases that were not even planned. The identity match between customers and company mergers can create a positive environment that strengthens customer confidence in impulse buying. In the image as an identity platform, the user interface has a significant role in purchasing decisions considering the aesthetic side.

**H6 is rejected** (sig. 0.104 < 0.05, β = 0.148), which means that Customer Perception has no significant effect on Brand Loyalty. The result is similar to the study from [103]. This shows that even though customers are loyal to the brand before it is merged, it does not guarantee that customers will remain loyal to the Merger. The Merger impacts the image of both companies, and customer perceptions may change due to a merger [5]. Thus, it can impact customer loyalty to the brand. Usually, customers judge the suitability of a merger from the valence of the brand (positive or negative) [2]. Therefore, companies need to choose whom they merge with and, as soon as possible, investigate customer responses to the Merger that will be carried out.

**H7 is accepted** (sig. 0.026 < 0.05, β = 0.261), which means that Brand Loyalty significantly affects Impulse Buying. The results in line with the study from [86] found a positive impact between brand loyalty and impulse buying. The results of this study indicate that loyal customers tend to do impulse buying even though the brand has been merged. This result has confirmed our findings from the FGD that has been carried out, where customers who remain loyal even though the two brands have merged will buy a product without planning if there is a stimulus in the form of product discounts or free shipping.

**H8 is accepted** (sig. 0.000 < 0.05, β = 0.390) and revealed that impulse buying significantly affects Purchase Intention. This finding supports the previous study by [90]. This paper has succeeded in bridging the gap between Customer perception on merger schemes and Purchase Intention through Customer Self-Congruency and Impulse Buying. Both have a close and inseparable relationship to the success of the role as a mediator. When customers are successfully given a stimulus that encourages impulse buying, purchase intention will certainly increase linearly. Impulse buying reduces the lines of consideration that customers think about before making a purchase decision, so impulse buying is an effective strategy in accelerating the buying process by customers.

Departing from the findings, that is needed to dig in-depth understanding of the effect of customer perception on merger scheme whether it gives a positive or negative effect on the company. Based on the comparative analysis based on the mean calculation of CP, the mean value is 5.67 with SD = 1.08. It is considered a positive effect since the mean value is higher than the median (4.00). The result concludes that the merger scheme of X and Y company is successfully launched with positive effect.

Besides, the additional analysis tries to compare the effect of the platform's existence or position in the market. Comparative analysis based on the mean calculation

of Customer Self-Congruency, Impulse Buying, Brand Loyalty, and Purchase Intention. In the CSC context, X platform has higher congruency than Y platform in the market (X:5.15 (SD = 1.00) > 4.88 (1.15)). All indicators of each variable are included in this analysis. In the context of Impulse Buying, the X platform has a higher impulse buying value than the Y platform (X:3.76 (SD = 1.61) > 3.61 (1.66)). In the BL context, BL value of Y platform higher than X platform (X:5.33 (SD = 1.14) > 5.06 (1.33)). The last, in the PI context, the X platform has a higher PI value than the Y platform (X:5.03 (SD = 1.31) > 4.80 (1.33)).

In summary, platform X is superior in building CSC, IB, and PI. However, BL is superior in the Y platform (see Figure 3). However, the differences that appear in each context are not significant, with a slight deviation.

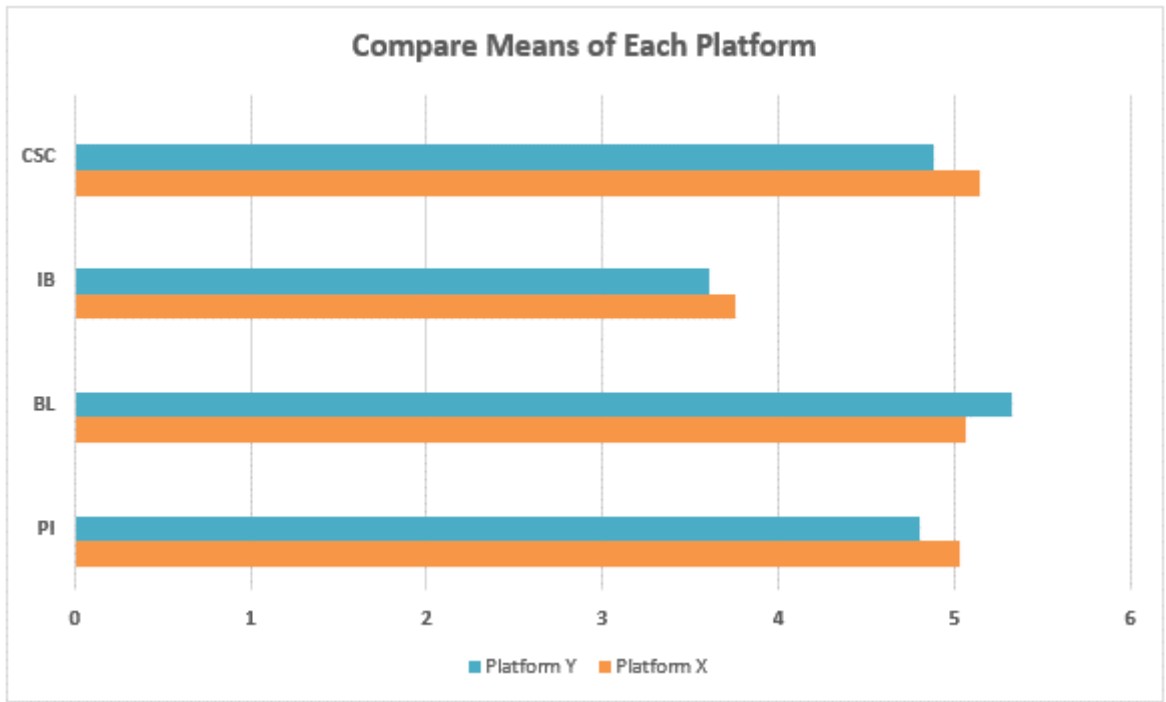

**Figure 3.** Compare Means of Each Platform. Source: Own Study.

### 5.1. Theoretical Contribution

This study provides several theoretical contributions as follows; first, the analysis carried out succeeded in closing the contradictory gap between Customer Perceptions on merger schemes on Purchase Intention through Customer Self-Congruency and Impulse Buying. This finding fills the body of knowledge in the domain of customer behavior research. Second, the study successfully overcame the theory application void by conducting a study related to customer perceptions of the merger scheme for decacorn companies that has never been done in Indonesia. It gives a depth understanding of where customer perceptions of the Merger of two decacorn companies cannot guarantee that customers will be loyal to the company. Third, this study provides new insight by measuring the impact of customer perceptions on purchase intention through the separation of indicators on X and Y companies so that this study can explain in detail the company's role in building each construct in the context of a merger.

### 5.2. Implications for Practice

Based on the above discussion, there are several implications for practice that can be considered recommendations in decision making as follows, first, building a company image through the user interface platform is crucial. The user interface is one of the company's image representations, which has proven to play an essential role in building customer self-congruence. The first aspect that is seen in the company is the user interface

itself. Thus, companies can provide regular evaluations of the user interface (is it user-friendly for customers? Is the aesthetic displayed in line with the character of the target customer?).

Second, because the user interface is the focus of customers and is visible on the platform's main page, marketers are recommended to ensure that the platform page is informative and aesthetic. Marketers can also add a stimulus that can invite interest from customers, for example, a change in the theme of the user interface at certain events, whether national events such as the new year or custom from the company. Based on the results of the FGD, it was also found that customers became disloyal after changes to the confusing user interface. This problem must be addressed immediately by educating customers regarding the use of the platform. Education will not work if given formally and monotonously, and it needs an element of creativity such as adding challenge features or games with prizes, which indirectly make customers move from one menu to another as a form of feature recognition and create habits.

Third, for companies, especially those engaged in e-commerce and technology-based services, the merger scheme can be one of the marketing strategies that can be a stimulus to customers to make impulse buying, which impacts increasing purchase intention. Merger schemes can improve the company's performance but by paying attention to the partners to have collaborated. The position and image of each company have a significant influence on the success of the merger scheme. In this study, company X tends to be stronger in positively impacting the merger scheme. This inequality is also suspected to be the cause of the ineffectiveness of customer perceptions on brand loyalty and customer self-congruency on purchase intention, so that in conducting a merger, it is necessary to ascertain the factors that need to be improved. For instance, a company with a stronger position and image needs to formulate new strategies to increase customer loyalty. Meanwhile, companies with weaker positions and images need to focus on creating strategies to improve three components, namely customer self-alignment, impulse buying, and purchase intention. That could be triggered by increasing sales promotion in various channels and branding strategies by identifying the customer-brand associations.

From the results of qualitative analysis in this study, it was found that consumers have expectations for companies that are merging, including low-competitive prices, more discounts, promotions, and cashback, easy and friendly user interface, cooperate with more partners, so more product choices to compare, and the company is more concerned with the environment. Companies that conduct mergers are advised to facilitate expectations from consumers so that old consumers do not move to competitors and, on the other hand, also attract new consumers, for example, by informing their customers regarding the M&A and benefits that the customers can retain, also activating customer feedback survey to practice the open innovations with the customers.

## 6. Conclusions

Even though M&A is one of the strategic actions to enlarge the market bases, our study shows that customer perception is not as simple as combining two brands. This paper has succeeded in bridging the gap in an inconclusive study between customer perceptions of the merger scheme and purchase intention. This paper offers customer self-congruency and impulse buying as mediating roles. Customer perceptions of the merger scheme at X and Y companies have an impact on several aspects, such as the emergence of a new identity harmony between customers and the company's image in terms of the user interface built on the platform and affects the purchase decision making process of customers as measured by purchase intention. However, departing from the model built in the study, customer perception of the merger scheme cannot influence customer loyalty to the company. That is because customers still have other factors to consider in deciding to be loyal or not. This paper proves that in increasing brand loyalty, customer perceptions must be directed first at the formation of self-congruency in the corporate image to gain trust and build robust engagement.

Like other studies, this research is also inseparable from limitations. First, the data used in this paper has met the minimum requirements for testing; however, to ensure the generalizability of research results, it is recommended to the extent the total sample with diverse respondents to cover all customer characteristics such as technology readiness, customer spending pattern, and habits. Second, this study limits the observations on brand loyalty. It is possible to extend the study to cover the gap of the brands' variables by examining other factors such as product type, brand image, and brand trust. Besides, there is also a need for further research regarding the gap between customer self-congruence on purchase intention except impulse buying, to see another mediation possibility. Third, this research is limited to only researching technology-based startup industries, which are fast-paced in nature and more customer or user-oriented (i.e., business-to-customer). The results may vary across industries. Future research can examine customer responses to M&A from different industries, such as finance, agriculture, or biotech industries that might have different industry characteristics, to examine the consistency of the results. Finally, data collection in this study used a cross-sectional method which this method could not describe whether there was a change in customer perceptions of company mergers over time. Future research can use the longitudinal method to find out whether the customer perception changes over time.

**Author Contributions:** Conceptualization, I.F.P. and G.H.; methodology, I.F.P., P.F.B. and G.H.; formal analysis, I.F.P. and G.H.; investigation, I.F.P. and G.H.; resources, I.F.P. and G.H.; data curation, I.F.P. and G.H..; writing—original draft preparation, I.F.P. and G.H.; writing—review and editing, I.F.P., N.A.W., G.H. and P.F.B.; visualization, I.F.P. and G.H.; supervision, N.A.W. and P.F.B. All authors have read and agreed to the published version of the manuscript.

**Funding:** This research received no external funding.

**Institutional Review Board Statement:** Not applicable.

**Informed Consent Statement:** Not applicable.

**Data Availability Statement:** Not applicable.

**Conflicts of Interest:** The authors declare no conflict of interest.

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
