# Peer review of "IS TWO ALWAYS BETTER THAN ONE? Customer Perception on the Merger of Startup Decacorn Companies"

_2199-8531, doi:10.3390/joitmc7040239_

Round 1

Reviewer 1 Report

The authors should consider the following recommendations in order to improve the original manuscript:

- To include the structure of the paper in the Introduction section.

- Authors should take into consideration much more recent publications in the sphere of discussed subject matter, especially studies conducted during the last 5 years. Regarding consumer behaviour framework but also on socioeconomic, demographic and ideological differences new perspectives, includind the impact of COVID-19 pandemic, I suggest extending the literature section by including at least the following relevant studies:

  1. Hawaldar, I.T.; Ullal, M.S.; Birau, F.R.; Spulbar, C.M. Trapping Fake Discounts as Drivers of Real Revenues and Their Impact on Consumer’s Behavior in India: A Case Study. Sustainability 2019, 11, 4637.
  2. Bonera, M.; Corvi, E.; Codini, A.P.; Ma, R. Does Nationality Matter in Eco-Behaviour? Sustainability 2017, 9, 1694.
  3. Batool, M., Ghulam, H., Hayat, M.A., Naeem, M.Z., Ejaz, A., Imran, Z.A., Spulbar, C., Birau, R. & Gorun, T.H. (2020) How COVID-19 has shaken the sharing economy? An analysis using Google trends data, Economic Research-Ekonomska Istraživanja, https://www.tandfonline.com/doi/full/10.1080/1331677X.2020.1863830.
  4. Antonides, G. Sustainable Consumer Behaviour: A Collection of Empirical Studies. Sustainability 2017, 9, 1686.
  5. Ullal, M.S., Spulbar, C., Hawaldar, I.T., Popescu, V. & Birau, R. (2021) The impact of online reviews on e-commerce sales in India: a case study, Economic Research-Ekonomska Istraživanja, DOI: 10.1080/1331677X.2020.1865179, https://www.tandfonline.com/doi/full/10.1080/1331677X.2020.1865179.

- Deepen the description of the limitations of conducted research and indicate the trends for further empirical research.

  • To expand the managerial implications in the article.
  • The sources must be added under each table and figure.
  • The Conclusions section is missing which is not acceptable.
  • Human proofreading, English grammar and spelling correction are also required in order to improve the quality of the manuscript.
  • I would also like to see a well-developed discussion comparing and contrasting solution/results presented in the work with existing work and then a subsection of it presenting contributions to theory/knowledge/literature and followed by a subsection on “Implications for practice”.

Author Response

Response to Reviewer 1 Comments

Point 1: To include the structure of the paper in the Introduction section.

Response 1: Following the comments from the reviewer, we have included the structure of the paper in the introduction section, which can be seen at L77-L84.

Point 2: Authors should take into consideration much more recent publications in the sphere of discussed subject matter, especially studies conducted during the last 5 years. Regarding consumer behaviour framework but also on socioeconomic, demographic and ideological differences new perspectives, includind the impact of COVID-19 pandemic, I suggest extending the literature section by including at least the following relevant studies:

  1. Hawaldar, I.T.; Ullal, M.S.; Birau, F.R.; Spulbar, C.M. Trapping Fake Discounts as Drivers of Real Revenues and Their Impact on Consumer’s Behavior in India: A Case Study. Sustainability 2019, 11, 4637.
  2. Bonera, M.; Corvi, E.; Codini, A.P.; Ma, R. Does Nationality Matter in Eco-Behaviour? Sustainability 2017, 9, 1694.
  3. Batool, M., Ghulam, H., Hayat, M.A., Naeem, M.Z., Ejaz, A., Imran, Z.A., Spulbar, C., Birau, R. & Gorun, T.H. (2020) How COVID-19 has shaken the sharing economy? An analysis using Google trends data, Economic Research-Ekonomska Istraživanja, https://www.tandfonline.com/doi/full/10.1080/1331677X.2020.1863830.
  4. Antonides, G. Sustainable Consumer Behaviour: A Collection of Empirical Studies. Sustainability 2017, 9, 1686.
  5. Ullal, M.S., Spulbar, C., Hawaldar, I.T., Popescu, V. & Birau, R. (2021) The impact of online reviews on e-commerce sales in India: a case study, Economic Research-Ekonomska Istraživanja, DOI: 10.1080/1331677X.2020.1865179, https://www.tandfonline.com/doi/full/10.1080/1331677X.2020.1865179.

Response 2: Thank you for the suggestions. We have added some of the literature provided by reviewers that is relevant to this study, which are Hawaldar et al (2019) and Ullal et al (2021). Besides, we also follow reviewers’ suggestions to add several additional literatures in the recent 5 years particularly in the area of Open Innovation and Consumer Behavior on Merger and Acquisition to strengthen the novelty and positioning of this study, which are:

  1. Kumar et al (2017) and Turoń, & Kubik (2021) are related to the impact of COVID-19 Pandemic.
  2. Peñarroya-Farell & Miralles (2021), Robbins et al (2021), Chu et al (2019), Marzec & Sliż (2020), Valdez et al (2021), Diaz & Duque (2021), and Tajudeen et al (2019) are related to the Open Innovation.
  3. Kim & Kim (2020) are related to Purchase Intention.
  4. Yao & Wang (2018) and Gaustad et al (2018) are related to the hypothesis development of Customer Self-Congruency and Purchase Intention (H2).
  5.  

Point 3: Deepen the description of the limitations of conducted research and indicate the trends for further empirical research.

Response 3: We identify at least four limitations from our study, ranging from the characteristics of the sample profiles, variable used, our case on technology-based startup, and the cross-sectional data collection, which might limit our results. We have added the limitations in the revised manuscript and indicate the trends for future research, which can be seen at L678-L695.

Point 4: To expand the managerial implications in the article.

Response 4: Following the comments from the reviewer, we have expanded the managerial implications of this research by providing more elaborations on how to meet customer expectations by applying open innovation. which can be seen at L618-L662.

Point 5: The sources must be added under each table and figure.

Response 5: In line with the comments from the reviewer, we have added sources under each table and figure.

Point 6: The Conclusions section is missing which is not acceptable.

Response 6: The conclusion section has been added, can be seen at L663-695.

Point 7: Human proofreading, English grammar and spelling correction are also required in order to improve the quality of the manuscript.

Response 7: We have fixed the grammar.

Point 8: I would also like to see a well-developed discussion comparing and contrasting solution/results presented in the work with existing work and then a subsection of it presenting contributions to theory/knowledge/literature and followed by a subsection on “Implications for practice”.

Response 8: Thank you so much for the constructive feedback. We agree that it is important to do benchmarking with previous studies, to see how it differs and supports the current knowledge. Our discussion shows that H1 is accepted, which means that customer perceptions on merger scheme has a significant effect on customer self-congruency, in line with study from Chung, Y., & Kim, A. J. (2020), Shamahet et al (2018), and Sung & Huddleston (2018). H2 is rejected, which means that customer self-congruency has no significant effect on purchase intention. These findings contrast with the study from Lee & Lee (2015) however similar with study from Pradhan et al (2016) and Phua & Kim (2018). H3 is accepted, which means that customer self-congruency has a significant effect on brand loyalty, in line with study from Han& Choi (2019). H4 is accepted, which means that brand loyalty has a significant effect on purchase intention. This finding is in line with study from Calvo et al (2016). H5 is accepted, which means that customer self-congruency has a significant effect on impulse buying, in line with study from Naeem (2021) and Japutra et al (2019). H6 is rejected, which means that Customer Perceptions has no significant effect on Brand Loyalty. This finding similar with the study from Thozhur et al (2007). H7 is accepted, which means that Brand Loyalty has a significant effect on Impulse Buying. This results in line with the study from Rashid et al (2019). H8 is accepted, which means that impulse buying has a significant effect on Purchase Intention. This finding supports previous study by Halim et al (2020).

The discussion on comparing the results of this study with previous studies has been added on L504 - L581. Furthermore, we have contributed to the theory subsection and implications for practice subsection, which can be seen in L605-L662.

Reviewer 2 Report

Dear author(s), thanks for your interesting paper devoted to the consumer perception of decacorn companies in the Indonesian environment. Despite the fact, the study is rather interesting, my overall assessment will not be that positive. The study needs to be revised significantly before publishing.

First of all, there are many mistakes, incorrect spelling, grammar tenses are not used properly, which makes it difficult to understand the text properly. Thus, the proofreading of the paper is a need. 

Introduction - it should be clearly stated, why a one country study is important for international readers. Please, indicate also the theoretical and practical contributions of the study.

Material and Methods - This section is a mixture of the data description and literature review. Study 1 (qualitative study) is based on a sample of 5 respondents with very similar demographic features, considering their age (can these findings be generalized? was the sample representative?)

L153 -L315 - it is neither the material nor the method - it is the development of the hypotheses (Literature review), should be placed somewhere else. 

The description of methods used is completely missing (Cronbach´s alfa, regression model- assumptions of the regression model were tested? multicollinearity is discussed, but what about the autocorrelation of residuals, etc,?)

Discussion - the results have to be discussed and compared in the context of other studies published worldwide - are they similar? different? this is missing in the current discussion.

Please, add the limitations of your study into the Conclusions. 

Author Response

Response to Reviewer 2 Comments

Point 1: First of all, there are many mistakes, incorrect spelling, grammar tenses are not used properly, which makes it difficult to understand the text properly. Thus, the proofreading of the paper is a need.

Response 1: We have fixed the grammar.

Point 2: Introduction - it should be clearly stated, why a one country study is important for international readers. Please, indicate also the theoretical and practical contributions of the study.

Response 2: Thank you so much for the feedback. We acknowledge that it is important to justify how this one country case affect and important for the global market, especially on tech-based start-ups. We use Indonesia data has the biggest customer base for online consumers, with its 88.1% of internet users in Indonesia (or nearly 240 million peoples) make purchases online. This makes Indonesia ranked first online consumers globally that become the customer base of e-commerce and tech start-ups across the globe [74]. Furthermore, considering the growth and number of start-ups, Indonesia currently occupies the 5th position globally with 2,193 start-ups companies [1]. Therefore, considering those reasons, Indonesia dataset is relevant and representative for the studies of market and start-ups decacorns.

Point 3: Material and Methods - This section is a mixture of the data description and literature review. Study 1 (qualitative study) is based on a sample of 5 respondents with very similar demographic features, considering their age (can these findings be generalized? was the sample representative?)

Response 3: We choose respondents with an age range of 20 since based on the reference, majority of consumers who use the services of these two companies are within that age range (Haryanto, 2017; Lidwina, 2021). Therefore, in terms of studying the customer perception would be representative.

However, we realize that it possibly limits the findings, especially if the context would be extended into another type of start-ups. Therefore, in the limitation section, we recommend extending the total sample with diverse respondents to cover all customer characteristics such as technology readiness, customer spending pattern and habits.

Point 4: L153 -L315 - it is neither the material nor the method - it is the development of the hypotheses (Literature review), should be placed somewhere else.

Response 4: In line with the comments from the reviewer, we have moved the hypotheses development into the literature review section (L194 – L332).

Point 5: The description of methods used is completely missing (Cronbach´s alfa, regression model- assumptions of the regression model were tested? multicollinearity is discussed, but what about the autocorrelation of residuals, etc,?)

Response 5: Thank you for the comments. In the revised version, we re-run and prove our hypothesis testing using PLS-SEM. The results are consistent and more reliable in showing the model fit, including measurement model, outer loadings, reliability, convergent validity, and discriminant validity. Considering the model, reflective constructs used in this study. The results of the analysis can be seen at L430 – L492.

Point 6: Discussion - the results have to be discussed and compared in the context of other studies published worldwide - are they similar? different? this is missing in the current discussion.

Response 6: Comparisons between the results of this study and previous studies have been added to the discussion session. Our comparison shows that We agree that it is important to do benchmarking with previous studies, to see how it differs and supports the current knowledge. Our discussion shows that H1 is accepted, which means that customer perceptions on merger scheme has a significant effect on customer self-congruency, in line with study from Chung, Y., & Kim, A. J. (2020), Shamahet et al (2018), and Sung & Huddleston (2018). H2 is rejected, which means that customer self-congruency has no significant effect on purchase intention. These findings contrast with the study from Lee & Lee (2015) however similar with study from Pradhan et al (2016) and Phua & Kim (2018). H3 is accepted, which means that customer self-congruency has a significant effect on brand loyalty, in line with study from Han& Choi (2019). H4 is accepted, which means that brand loyalty has a significant effect on purchase intention. This finding is in line with study from Calvo et al (2016). H5 is accepted, which means that customer self-congruency has a significant effect on impulse buying, in line with study from Naeem (2021) and Japutra et al (2019). H6 is rejected, which means that Customer Perceptions has no significant effect on Brand Loyalty. This finding similar with the study from Thozhur et al (2007). H7 is accepted, which means that Brand Loyalty has a significant effect on Impulse Buying. This results in line with the study from Rashid et al (2019). H8 is accepted, which means that impulse buying has a significant effect on Purchase Intention. This finding supports previous study by Halim et al (2020).

For more details, see (L504 - L581).

Point 7: Please, add the limitations of your study into the Conclusions.

Response 7: The conclusion and limitation of this study has been added in section 6, namely the conclusion (can be seen in L678 – L695).

Reviewer 3 Report

The topic of the paper is of great interest. The influence of company merging on customers’ perception is not well-researched topic. The advantage of the manuscript is using a curious case. But it is completely incomprehensible how the conceptual framework used is related to the merger of companies. Not in the scheme, not in the hypotheses, there is nothing about it. By itself, the scheme used is rather trivial. That hypothesis and schema are related to the merger are clear only from the questions on pages 10-11. It is necessary to include reasoning about the merger much earlier, at least take into account the merger when describing hypotheses.
p. 10 - typos in the table
table 8 - a description of the sample should be provided in the methods
figure 3 - decipher which indicators are presented 

Author Response

Response to Reviewer 3 Comments

Point 1: The topic of the paper is of great interest. The influence of company merging on customers’ perception is not well-researched topic. The advantage of the manuscript is using a curious case. But it is completely incomprehensible how the conceptual framework used is related to the merger of companies. Not in the scheme, not in the hypotheses, there is nothing about it. By itself, the scheme used is rather trivial. That hypothesis and schema are related to the merger are clear only from the questions on pages 10-11. It is necessary to include reasoning about the merger much earlier, at least take into account the merger when describing hypotheses.

Response 1: Thank you for the comments. In line with the comments from reviewers to add literature related to Merger when explaining the development hypothesis, we have added some literature from (Yao & Xinxin, 2018) explaining merger and cquisition (M&A) carried out by the company will impact the image and brand identity. (Gaustad et., 2018) explaining due to M&A, a brand's identity may take on undesirable traits; for example, customers' re-flections of their self- and ideal image through a brand may be distorted. (Thorbjørnsen & Dahlén 2011) explaining if consumers have a negative perception of a merger and acquisition (M&A) brand, then it can affect customer loyalty and have a greater intention to switch brands. (Legendre & Bowen, 2020) explaining that it is vital to explore how customers deal with M&A identity dissonance induced by changes in brand meanings to produce more successful M&A results and better secure existing customers. (Dezi et al., 2018) explaining companies frequently employ mergers and acquisitions to improve their performance and attract customers, and (Kato & Schoenberg, 2012) explaining during the post-merger acquisition integration stage, customers' purchasing intentions were directly influenced by perceived changes in service quality and product variety. We add literature to clarify the relevance between the hypotheses development with the Merger and Acquisition.

Point 2: p. 10 - typos in the table

Response 2: Thank you for the comment. We have fixed it.

Point 3: table 8 - a description of the sample should be provided in the methods.

Response 3: The table and description of the respondents in this study have been moved according to advice from reviewers, namely to the methods section. It can be seen in L407-415.

Point 4: figure 3 - decipher which indicators are presented.

Response 4: Comparative analysis based on the mean calculation of Customer Self-Congruency, Impulse Buying, Brand Loyalty, and Purchase Intention. In the CSC context, X platform has higher congruency than Y platform in the market (?Ì…: 5.15 (SD=0.99) > 4.88 (1.15)). All indicators of each variable are included in this analysis.

Round 2

Reviewer 1 Report

The original manuscript has been significantly improved. The authors followed the recommendations included in the previous review report so that the quality of their research article has greatly increased.The revised version of the manuscript complies with the JOItmC journal standards. I also appreciate the hard effort of the authors in this regards.

Reviewer 2 Report

Dear author(s), your paper was improved significantly, following the comments and recommendations.